# Age-Related Male Hypogonadism and Cognitive Impairment in the Elderly: Focus on the Effects of Testosterone Replacement Therapy on Cognition

**DOI:** 10.3390/geriatrics5040076

**Published:** 2020-10-16

**Authors:** Giuseppe Lisco, Vito Angelo Giagulli, Anna De Tullio, Giovanni De Pergola, Edoardo Guastamacchia, Vincenzo Triggiani

**Affiliations:** 1Interdisciplinary Department of Medicine–Section of Internal Medicine, Geriatrics, Endocrinology and Rare Diseases. School of Medicine, University of Bari, Piazza Giulio Cesare 11, Policlinico of Bari, 70124 Bari, Italy; g.lisco84@gmail.com (G.L.); vitogiagulli58@gmail.com (V.A.G.); annadetullio16@gmail.com (A.D.T.); edoardo.guastamacchia@uniba.it (E.G.); 2ASL Brindisi, Unit of Endocrinology, Metabolic Disease & Clinical Nutrition, Hospital “A. Perrino”, Strada 7 per Mesagne, 72100 Brindisi, Italy; 3Outpatients Clinic of Endocrinology and Metabolic Disease, Hospital “F. Jaja”, Via Edmondo de Amicis 36, Conversano, 70014 Bari, Italy; 4Department of Biomedical Sciences and Human Oncology, Section of Internal Medicine and Clinical Oncology, University of Bari Aldo Moro, Piazza Giulio Cesare 11, 70124 Bari, Italy; giovanni.depergola@uniba.it

**Keywords:** late-onset hypogonadism, testosterone, testosterone replacement therapy, cognition, memory, mild cognitive impairment, review

## Abstract

**Background.** Epidemiological data report that male hypogonadism may play a role in cognitive impairment in elderly. However, the effect of testosterone replacement therapy (TRT) on cognitive abilities in this cluster of patients has not been well established. **Methods.** PubMed/MEDLINE, Google Scholar, Cochrane Library, and Web of Science were searched by using free text words and medical subject headings terms related with “male hypogonadism”, “late-onset hypogonadism”, elderly, cognition, “mild cognitive impairment”, memory, “testosterone replacement therapy” used in various combinations according to the specific clinical questions. Original articles, reviews, and randomized controlled trials written in English were selected. **Results.** A long-term TRT could improve specific cognitive functions, such as verbal and spatial memory, cognitive flexibility, and physical vitality. However, randomized controlled trials do not provide positive results, and in most of the cases TRT might not induce beneficial effects on cognitive function in elderly men. **Discussion and conclusions.** Since the lengthening of life expectancy, the prevalence rate of cognitive decline in elderly men is expected to increase remarkably over the next decade with considerable healthcare and economical concerns. Therefore, this remains a relevant clinical topic and further investigations are needed for clarifying the role of TRT especially in elderly men with hypogonadism.

## 1. Aging-Related Hypogonadism in Men

The term late-onset hypogonadism (LOH) has been introduced to identify a syndrome characterized by both clinical and biochemical evidence of low concentrations of testosterone (T) associated with advancing age in males [1]. Specifically, an aging-related decline in gonadal function has been observed starting from 35 years of age with a more marked decline in the serum concentrations of T after the 7th decade of life [2]. This phenomenon is attributable to several mechanisms, including a progressive decline in the number of normally functioning Leydig cells in aged testicles, lower testicular response to luteinizing hormone (LH), along with a gradual increase in serum concentration of the sex-hormone-binding globulin (SHBG) with a consequent reduction in both serum free and bioavailable T concentrations. In most of the cases, LOH might be the consequence of the use of medications (glucocorticoids, opioid analgesics, antidepressants, or aldosterone antagonists) or chronic diseases [3,4,5,6,7,8,9]. The age-related decline in serum T concentrations has been described in both longitudinal (−1.6%/year) and cross-sectional studies (−0.8%/year) [10]. According to the results of the European Male Aging Study, LOH was diagnosed in 23.3% of participants (mean age 60 years) [11].

## 2. Diagnostic Challenges of Male Hypogonadism in the Elderly

A comparative analysis of evidence-based recommendations and guidelines highlighted a significative grade of discordance, thus generating confusion about different issues [12]. First, the best age-specific cutoff value to define the onset of LOH remains equivocal. Moreover, signs and symptoms suggestive of male hypogonadism are heterogeneous, and guidelines and recommendations do not always categorize the level of specificity of these clinical markers, making the diagnostic workup dissimilar in comparable clinical contexts especially in the elderly. Finally, the goal of an optimal T replacement therapy is dissimilar and is often not appropriate for aged men. A routinely assessment of gonadal status in aged men is not recommended, and testing appears appropriate only in the case of highly suspicious patients such as those with sexual dysfunction [13,14]. However, sexual dysfunction is a very common finding in the elderly and might be related with underlying conditions other than male hypogonadism. Moreover, poor specific signs and symptoms, such as cognitive or mood change, unexplained anemia, loss of muscle bulk and strength, increase of visceral abdominal fat, fatigue, and social isolation, can be frequently associated with male hypogonadism, especially in the elderly, but they are often attributable to other common conditions in this clinical setting, hence, leading to misdiagnosis.

## 3. Cognitive Impairment in the Elderly

According to the Center for Disease Control and Prevention (https://www.cdc.gov/aging/aginginfo/subjective-cognitive-decline-brief.html), the prevalence of subjective cognitive decline is 11%, more in men (11.3%) than in women (10.6%) and among the elderly (>65y: 11.7%) respective to youngers (45–59y: 10.8%). Cognitive decline may be considered a risk factor for the future development of dementia [15], but it is usually the result of a physiological cognitive decline associated with aging (cognitive aging), especially in community-dwelling elderly [16]. Patients with a cognitive impairment experienced a decline in several cerebral functions, which are normally necessary to maintain normal daily life routines and daily activities, including memory, language, thinking, and judgment. The so called “mild cognitive impairment” (MCI) is a clinical stage comprised between cognitive aging and a mild dementia and is characterized by a subjective and objective functional decline, which includes memory complaints and abnormal memory for age but without any evidence of decline in cognition, impairment of daily life activities, and dementia [17]. Its estimated that prevalence ranges from 2.5 per 1000 person-year for ages 75–79 years to 60.1 per 1000 person-year for ages > 85 years [18]. Cognitive impairment over time is related with several clinical conditions, which include genetic predisposition, geographical distribution, cardiovascular diseases, education, cigarette smoking, poor physical activities, diabetes mellitus, oxidative-inflammatory status, arterial hypertension, atherosclerosis, atrial fibrillation, vitamin D and vitamin B insufficiency or deficiency, sleep-breathing disorders, longer duration of sleep, metabolic syndrome and type 2 diabetes, psychiatric or neurodegenerative illness, brain injury, and hormonal imbalance (including male hypogonadism) [19,20,21,22,23,24,25,26,27,28,29,30,31,32,33,34,35].

## 4. Testosterone Deficiency and Cognition

Androgen receptors have been found in several cerebral areas, such as in the medial preoptic area, ventro-medial hypothalamus, medial amygdala, nucleus accumbens, stria terminalis end septum, and cerebral cortex [36]. Androgen metabolism is particularly active in the brain; T and its active metabolites, such as estradiol and dihydrotestosterone (DHT), play a role in the regulation of brain development [37]. Sexual steroids are known to modulate endothelial function, thus regulating blood flow and vascular inflammation also in brain [38]. In Alzheimer’s disease, an overproduction of amylogenic precursors, such as amyloid beta-peptide, and their accumulation in several areas of brain is involved in the pathophysiology of neurodegeneration [39,40]. T was found to reduce the secretion of amyloid beta-peptide and increase the secretion of the non-amyloidogenic fragment, sbetaAPPalpha, from mouse neuroblastoma cells line (N2a) and rat primary cortico-cerebral neurons, thus demonstrating to positively modulate the intracellular metabolism and protect against Alzheimer’s disease evolution [41]. This protective effect exerted by androgens, including T and its metabolites, has been confirmed by other observations [42,43,44,45]. Additionally, patients with Klinefelter syndrome usually display worse verbal processing speed and verbal execution function both associated with an imaging evidence of reduced brain left temporal lobe volume [46]. These neurocognitive outcomes were assumed to be related not only with supernumerary X chromosomes but also with lower concentrations of serum T, frequently observed in this cluster of patients [46]. Moreover, different psychiatric disorders such as schizophrenia, autism, attention-deficit hyperactivity disorder, depression, and anxiety have been frequently reported in individuals with Klinefelter syndrome and testosterone replacement therapy (TRT) usually fails to restore these disorders even when it is prescribed for correcting the underlying hypogonadism [47].

A longitudinal study reported a relationship between T decline over time and decline in cognitive function in men [48,49]. To confirm a tight correlation between low concentrations of T and cognitive decline, a propensity-score matching analysis demonstrated that the risk of dementia was higher among patients who were prescribed 5 alpha-reductase inhibitors [50]. T decline may affect cognitive processing speed in elderly men even in the case of mild or compensated hypogonadism [51]. Working memory, which is considered as the ability to acquire and maintain information and flexibly manage or update it, should be considered a critical gateway for a broad range of cognitive skills and is normally affected by sexual hormone imbalance in both of genders [52]. Conversely, memory, learning capacity, and cognitive activity are positively modulated by T and its active metabolites as suggested by the evidence of the higher prevalence of these dysfunctions in hypogonadal men [53]. Recent evidence reported that elderly men with hypogonadism are at higher risk of developing neurodegenerative dementia [54]. Given these considerations, TRT could be useful for improving cognition in elderly patients. Despite these evidences, the importance of cognitive and mood declines over time, especially in aged men, could be clinically underestimated. Indeed, not all guidelines considered cognitive impairment as a suggestive item to be considered for diagnosing male hypogonadism and, even if well recognized, neurocognitive signs and symptoms are normally considered as mildly or moderately suggestive for male hypogonadism, especially for LOH [12]. To settle the diagnostic challenges of LOH, validated questionnaires may be useful, but even in this case only a few of these tools considered cognitive symptoms as relevant clinical items, usually exploring favorable mood symptoms (Androgen Deficiency in Aging Males; the Massachusetts Male Aging Study questionnaire; Androtest) [55,56,57,58].

## 5. Testosterone Replacement Therapy and Cognition

PubMed/MEDLINE, Google Scholar, Cochrane library, and Web of Science were searched by using free text words and medical subject headings terms related with “male hypogonadism”, “late-onset hypogonadism”, elderly, cognition, “mild cognitive impairment”, memory, “testosterone”, “testosterone replacement therapy” used in various combinations according to the specific clinical questions. Original articles, reviews, and randomized controlled trials written in English only and published from 2000 to 2020, were selected. Additional filters included male sex and age > 65 years. Randomized controlled trials were selected specifically comparing TRT to placebo. The research focalized about the effect of TRT on age-related cognitive decline and MCI. Several randomized controlled trials have been carried-out in order to explore the effects of TRT on different outcomes related to cognitive functions (Table 1). Results were normalized on the basis of the education levels of participants. Study designs were different, particularly referring to the age of participants, gonadal status at baseline, and different cognitive impairment at baseline. Moreover, TRT supplementation were differently administered (oral, transdermal, intramuscular) and the exposure to TRT was variable according to the different follow-up strategies. In addition, T supplementations were aimed to achieve and maintain different serum T objectives (low-normal, normal, above the normal range), and methods of assay (radio- and chemiluminescence immune assay; liquid chromatography; liquid chromatography–mass spectrometry) as well as timing of serum T concentrations assessment (random or in the morning) were dissimilar among the different trials. Neurocognitive testing was carried-out, aiming to explore several cerebral areas (such as frontal, pre-frontal, temporal cerebral cortex) involved in the control of memory, verbal fluency, and visuo-spatial recognition.

In the Cognitive Functional Trial, 498 patients with an age-associated memory impairment (mean age 72 years; mean serum T concentrations at baseline 242 ng/dL) were randomized (1:1) to receive either T supplementation (gel formulation) aiming to obtain and maintain T concentrations in the normal range for young men or placebo. Even though the subgroup treated with TRT achieved higher concentrations of T compared to those on placebo, visual memory, executive function, and spatial ability did not improve at 6 and 12 months of treatment between the two groups [59]. Similar data were also obtained by the TEAAM trial in which 280 men with LOH and mild cognitive impairment were randomized to receive TRT versus placebo for 36 months. Several aspects of cognition were assessed and compared between the two groups, such as visual spatial ability, verbal fluency, verbal memory, manual dexterity, attention, and executive function after the adjustment for age, baseline cognitive function, and education [60]. Wahjoepramono et al. recruited and analyzed 50 healthy patients with normal cognitive function at baseline (age > 50 years) and a low-to-normal serum T concentrations, founding that TRT compared to placebo lead to a slight improvement of Mini-Mental State Examination (MMSE) testing and lower depressive symptoms assessed by the Geriatric Depression Scale (GDS) [61]. Among hypogonadal men (>60 years) with MCI, TRT per se slightly improved recognition and memory performance and reduced depressive symptoms [64]. Cherrier et al. observed a slight but significative improvement in both verbal and spatial memory assessed in community-dwelling volunteer men supplemented with T enanthate 100 mg once-weekly administrated for 6 weeks compared to placebo [77]. Other trials found that TRT leads to a slight but significative improvement in spatial reasoning performance, physical activity, and vitality as well as in cognitive flexibility, spatial memory, and verbal memory [70,75,76]. T metabolites, such as DHT and estradiol, may contribute to these positive effects on some cognitive functions, particularly verbal memory [71].

Despite these results, other authors did not confirm similar findings irrespective of the baseline characteristics of recruited patients, including gonadal status at baseline, formulations used for either a T replacement or supplementation, serum concentrations of T during and after the treatment, period of exposure to T and follow-up strategies [62,63,66,69,72,73,74,77]. In addition, supraphysiologic T concentrations might prompt detrimental effects especially on verbal memory [67].

## 6. Discussion and Conclusions

Male hypogonadism may be considered as a risk factor for cognitive decline especially in the elderly, and it is attributable to the loss of several neuroprotective effects provided by T at the level of the central nervous system. Moreover, normal levels of serum T seems to prevent cardiometabolic and coagulative disturbances that might be considered as risk factors for cognitive impairment and MCI [79,80,81,82,83,84,85]. Despite these evidences, T supplementation—even in eugonadal men—normally failed to demonstrate protective effects [86,87,88], and the results of the examined trials confirmed this trend. To partially explain this phenomenon, it should be considered that study designs and protocols were different. First, the mean age of participants ranged from 61 years [62] to 80 years [74]; participants were more than 70 years old in some trials [59,63,67,69,74,76] and less than 70 years old in the remaining studies [60,61,65,66,67,69,70,71,72,76,77,78]. In one trial [64], young patients were also included (mean age 28 years). As known, elderly men are more prone to both a baseline cognitive impairment and lower levels of serum T. Indeed, the mean levels of serum T at baseline were lower than 300 ng/dL in trials recruiting older men [59,63,66,68,73,75] excepted for one of them [74]. Conversely, the mean levels of serum T in trials recruiting younger patients were greater than 300 ng/dL [60,61,64,65,67,69,70,71,72,76,77,78]. Patients with baseline cognition impairment were included in only three trials [59,63,73] that recruited patients with a higher mean age (>70 years), and those with subjective memory complaints were included in only one [62]. The remaining trials assessed the effect of TRT in patients with normal cognition levels for age and education in both elderly (mean age > 70 years) [66,68,75] and younger patients (mean age < 70 years). Considering that TRT is the most effective treatment for restoring normal levels of serum T in hypogonadal men, it is expected that T supplementation may be more effective in hypogonadal compared to in eugonadal men. For the same fundamental conditions, patients with worse cognition and poor cognitive reserve could benefit from TRT better than those with no cognitive decline or complaints. Despite these assumptions, the results of a clinical trial did not find positive effects of TRT on global cognitive function in elderly patients with cognitive impairment and male hypogonadism [59]. Indeed, Resnik et al. [59] treated similar patients with Testogel 1% 5 mg with further titration to achieve serum T concentrations of 500–800 ng/dL. Patients who were randomized to TRT obtained a moderate increase in serum T levels (500 ng/dL in mean) after 3 months and achieved the target until the 9th month. Even though male hypogonadism was effectively treated, no improvement in cognitive function were found compared to placebo. Cherrier et al. [63] in younger patients (mean age 70 years) with mild cognitive decline, but with normal baseline serum T concentrations (mean T = 308 ng/dL), failed to find cognitive improvement due to TRT. In this case, Testogel posology were strictly titrated (3 months) to achieve and maintain a moderate increase in serum T concentrations to 500–900 ng/dL. Serum T concentrations increased after 3 months and remained stable at 6 months from randomization (650 and 600 ng/dL, respectively). Similar findings were finally found by Kenny et al. [73] in elderly eugonadal men (mean levels of serum T at baseline of 410 ng/dL) with cognitive decline. Patients were started on T enanthate 200 mg i.m. every 3 weeks for 3 months. Serum T concentrations raised up to 1211 ng/dL through the follow-up, hence achieving levels that, to date, we would define as supraphysiologic for age (80 years). Another cluster of clinical trials focalized on elderly patients (mean age >70 years) with LOH but normal cognition at baseline [66,68,73]. Among these trials, only in one [75], T supplementation induced a mild improvement in physical activity and vitality as well as in cognitive flexibility. In this case, T was administered by means of a transdermal T patch, 2–2.5 g/day for 12 months, obtaining a moderate increase in the levels of serum T. Maki [66] and Vaughan [68] failed to obtain significative improvement in cognitive function. In the former trial, T enanthate 200 mg (i.m.) was administered once-weekly for 90 days in order to achieve supraphysiologic serum T concentrations (T = 249 ➔ 970 ng/dL); in the latter, T enanthate 200 mg was administered every two weeks for 3 years, with or without finasteride 5 mg/day, to achieve moderate increase in serum T levels (T = 286 ➔ 588 ng/dL). Finally, the remaining trials assessed the effect of T on cognitive function in younger men (<70 years) without baseline hypogonadism and without cognitive impairment or complaints [60,61,65,67,69,70,71,72,74,76]. Among these, positive findings were found only in 5 out of 10 trials [61,67,70,74,76] and consisted in an improvement in spatial memory, spatial ability, and verbal memory. In all of the cases, TRT was administered for significantly improving baseline serum T concentrations by using T gel 50 mg per day for 26 weeks [61] and T enanthate 100 mg (i.m) once a week alone ([67,74,76], Cherrier M.M. et al. 2007) or in combination with anastrozole 1 mg per day [70].

Despite differences in serum T concentrations at baseline and considering the different formulations administered to replace hypogonadism, neither moderate nor supraphysiologic increases in serum T concentrations were able to improve cognitive function in patients with mild cognitive impairment irrespective of LOH. Possibly, TRT requires a long period of time to efficiently restore cognitive function, as observed in a wide range of clinical trials in which a single shot of transdermal T [78], as well as a short follow-up period [62,64] failed to improve cognitive function. Conversely, younger patients with normal cognitive function and without LOH seems to exhibit a slight improvement of spatial abilities and memory. These ameliorations were specifically found in the case of a remarkable T supplementation, while they become negligible in the case of a mild supplementation by means of T gel 1%, T undecanoate 160 mg/d, T enanthate (50–600 mg/w) [60,65,69,71,72].

Considering these heterogenous study designs and protocols, these positive results on cognition could be the consequences of several biases. In a recently published metanalysis, Buskbjerg et al. extensively evaluated the effect of T supplementation on cognition, finding a great variability in the results [89]. Particularly, different sources of bias have been verified. A poor detailed description of the methods of randomizations and allocation concealment, blinding of participants and researchers, incomplete outcome data reporting, and pharmaceutical sponsoring were found as the main confounding factors. Authors highlighted that statistically significant effects of T supplementation were found among subgroups such as those evaluating eugonadal men at baseline, younger men, studies administering T by injections, non-sponsored trials, and studies assessing random instead of morning T. Despite these findings, the final effect of T supplementation on cognitive function was minimal and statistically insignificant.

Probably, T supplementation per se does not improve cognitive function or prevent cognitive decline according to the currently available evidences. In order to explain this result, several observations might be done. Study designs had poor statistical robustness having been conceived for assessing heterogeneous outcomes starting from a usually low number of participants and considering numerous biases in studies protocols. Moreover, in some of the cases, study population did not focalize on the specific issue of male hypogonadism in the elderly. Indeed, in most of the cases, participants had normal or low-normal serum T concentrations and, in this cluster of patients, it is expected that T supplementation (and not replacement) may induce poor or even detrimental systemic effects (including those on cognition). In addition, a few trials assessed the effect of TRT/T supplementation in elderly patients, and in most of the cases participants were younger and therefore less prone to improve cognitive function due to a greater cognitive reserve [90]. This clinical matter remains to be clarified also in light of current guidelines and recommendations that did not specifically established the cutoff of age for the diagnosis of LOH [12], hence emphasizing the need of an accurate diagnostic workup of male hypogonadism in the elderly (clinical practice) [91] and an adequate study population selection (clinical research) only among those with high probability of having hypogonadism (and consequently in which TRT can be prescribed effectively and safely). Therefore, more studies are needed to better explore the role of hypogonadism in elderly men and the impact of TRT in patients with LOH and cognitive decline. Different T formulations have been used for the supplementation in clinical trials, mostly transdermal or intramuscular, and different T targets have been proposed once TRT was started. According to the clinical practice, TRT in the elderly should be carefully started and managed in order to avoid a detrimental overexposure to T, as well as dangerous fluctuation of serum T concentrations [92], which are expected to induce side effects more frequently (especially on cognition). According to this point of view, T formulations with a short half-life should be more appropriate for treating elderly men [92], but this clinical consideration does not have a scientific robust support as head-to-head trials aimed to compare both the efficacy and safety of different T formulations are currently lacking.

Besides these considerations, it should be noted that serum T concentrations do not necessarily replicate the concentrations of T in the central nervous system. Martin et al. recently observed that serum and cerebrospinal fluid steroid (including T) concentrations were weakly correlated to each other [93], thus confirming a difference between peripheral and central concentrations of T. Hence, the improvement of serum T, while patients is started on TRT/T supplementation, could not methodologically be the best biomarker to use in clinical trials, and further studies might be considered also in light of this further variable. 

Pending further clarification in the field [94,95], the main indication for TRT prescription also in elderly men remains the treatment of suggestive symptoms of hypogonadism associated with unequivocally low concentrations of serum T, thus it should not be specifically the aim to improve neurocognitive functions or prevent cognitive decline also in aged men with hypogonadism [96]. However, since the rate of prevalence of cognitive impairment and dementia in elderly is expected to increase remarkably in the next decade, due to the lengthening of life expectancy, this clinical topic is relevant making urgent further study in the field necessary [97].

## Figures and Tables

**Table 1 geriatrics-05-00076-t001:** Summary of randomized controlled trials that investigated the effect of testosterone supplementation in hypogonadal and eugonadal men with and without baseline cognitive impairment.

Authors (Year)	Population	Intervention	End-Points	Main Results
Resnick et al. (2017) [59]	788 men (493 with AAMI); age > 65 years (mean 72 years); at least 2 morning T < 275 ng/dL; sexual dysfunction; AAMI; reduced physical function and vitality	Randomization (1:1) to T vs. P for 12 months (247 T; 246 P)	Mean change in delayed paragraph recall (0 to 50); visual memory (Benton Visual Retention Test: 0 to −26), executive function (−290 to 290), and spatial ability (−80 to 80).	No difference between the two groups at 6 and 12 months.
Huang et al. (2016) [60]	Age > 60 years (mean 67 years); T = 100–400 ng/dL; healthy men	Randomization (1:1) to T gel 1% 7.5 mg vs. P for 3 years (156 T vs. 153 P)	Mean change in visual ability (complex figure test); phonemic or category verbal fluence (phonemic and categorical fluency test); verbal memory (paragraph recall test); manual dexterity (Grooved Pegboard test); attention and executive function (Stroop Interference test).	No difference between the two group at 6, 18, and 36 months (both intention-to-treat and per-protocol analysis).
Wahjoepramono et al. (2016) [61]	Age > 50 years; T = 300–600 ng/dL; normal baseline cognition (MMSE > 24); no relevant co-morbidities; education > 6 years	Crossover study; 50 recruited patients; randomization to T cream 50 mg ➔ P (A) or P ➔ T cream 50 mg (B); 24 weeks of active treatment for each (T or P) with 4 weeks of washout (52 weeks cumulatively)	Mean change in MMSE; Rey Auditory Verbal Learning test; intermediate and delayed recall; GDS.	TRT induced a slight but significant amelioration of MMSE score. The amelioration occurred when patients treated with placebo were shifted to T (B) and persisted when those started with T were shifted to P (A). Depression symptoms followed the aforementioned trend. Other outcomes remained unchanged.
Asih et al. (2015) [62]	Mean age 61 years; subjective memory complaints; low-to-normal gonadal status	44 patients, randomization (1:1), with crossover, to transdermal T 50 mg (24 weeks) then converted to P (24 weeks) after 4 weeks of washout and vice versa	Mean difference in androgens and estradiol serum concentration; biomarkers of efficacy/safety of TRT (red blood cells count, hemoglobin, PSA; plasmatic insulin concentration; body mass index and body fat mass); plasma amyloid beta protein concentration (biomarkers of dementia).	No difference in concentrations of plasma amyloid-beta protein concentrations.
Cherrier et al. (2015) [63]	Age 60–90 years (mean 70.5 years); mild cognitive impairment (Peterson criteria); T < 300 ng/dL; mild-to-moderate urinary symptoms (AUA symptom score <19)	22 eligible patients (10 T vs. 12 P) for 6 months	Mean change in attention or spatial ability; spatial and visual memory test, verbal fluency and working memory.	No difference between the two groups at 3 and 6 months.
Borst et al. (2014) [64]	Age > 60 years (mean 67 years); total T < 300 ng/dL or bioavailable T < 70 ng/dL; MCI	60 patients randomized to (2 × 2 design): vehicle–placebo (16) or T enanthate–placebo (14) or T enanthate–finasteride (17) or vehicle–finasteride (13)	Mean change in depression symptoms (GDS); visual–spatial processing (the Trail Making Test A and B, and the Benton Judgment of Line Orientation); psychological well-being (Life Satisfaction Index A and B); recognition and memory performance (the Rey–Osterrieth complex figure test at 0, 5, and 30 min recall test).	T reduced depressive symptoms (−0.74 point on GDS), and improved memory (+2.87 on a 36-point scale of the Rey–Osterrieth complex figure test at 30 min).
Young et al. (2010) [65]	Healthy younger (*n* = 22; 25–35 years) and older (*n* = 62; 60–80 years, 68 years in mean); no sign of cognitive decline (MMSE < 25) or depression (GDS < 10); normal functional intelligence (WAIS-R > 8); Total T = 241–827 ng/dL	Double-blind randomization in 4 groups: 1. GnRH agonist + T gel; 2. GnRH agonist + T gel + aromatase inhibitor; 3. GnRH agonist alone (inducing hypogonadism); 4. Placebo only. Follow-up visits 10, with a cumulative period of observation of 12 weeks	Mean change in Trail Making Test A (only numbers), B (numbers and letters) and Self-Ordered Pointing Test (working memory); number of words generated in 1 min starting from a letter of alphabet (verbal fluency); Mental Rotation Task and Figure Discrimination Task (spatial cognition); Paragraph Recall Test (verbal memory).	Hormonal concentrations did not alter cognition. However, free T concentrations were found to be positively related to spatial cognition and estrogen concentrations were negatively related with working memory performance.
Emmelot-Vonk et al. (2008) [66]	Healthy men; age 60–80 years (mean 67 years); T concentrations < 395 ng/dL	237 men were finally eligible; randomization (1:1) to T undecanoate (Andriol Testocaps) 80 mg twice-daily vs. placebo for 6 months.	Mean change in functional mobility, cognitive function, bone mineral density, body composition and anthropometry, blood pressure, safety. Cognitive assessment: the Dutch version of the Rey Auditory Verbal Learning Test immediate and delated recall (verbal memory); Digit–Symbol Substitution test; Trail-Making test; Benton Judgment of Line Orientation test; Vandenberg and Kuse adaptation of the 3-Dimensional Shepard Mental Rotation test.	No change in cognitive function tests.
Maki et al. (2007) [67]	Normal cognition; age 66–86 years (mean 74 years); T > 240 ng/dL	50 patients, T enanthate 200 mg i.m. weekly for 90 days vs. placebo	Change in verbal learning and memory (primary end-point); short-term memory of geometric figures (Benton Visual Retention test); working memory (Digit Span, α -Span); speeded manual dexterity (Grooved Pegboard test); attention, visuomotor scanning, and cognitive flexibility (Trail-Making Test); mood (and 6) Positive and Negative Affect Schedule). Glucose uptake on brain (frontal and temporal areas) positron emission tomography was also carried out.	No improvements. Supraphysiologic T supplementation in elderly had a potentially detrimental effect on verbal memory.
Cherrier et al. (2007) [68]	50–90 years (mean 67 years); community-dwelling volunteer men; low-normal gonadal status; education 12–19 years; normal cognition for age	57 patients; randomized to T enanthate (50, 100 or 300 mg) weekly i.m. or placebo for 6 weeks	Mean change in verbal and spatial memory (Puget Sound Route Learning Test and Word List Recall).	Only patients who exhibited a moderate increase (402 ➔ 1184 ng/dL) in T concentrations compared to those achieved low improvement (366 ➔ 436 ng/dL) or high improvement (355 ➔ 3141 ng/dL), improved both verbal and spatial memory.
Vaughan et al. (2007) [69]	Healthy men; no evidence of cognitive impairment; age 65–83 years (mean 70 years); 2 morning serum T concentrations < 350 ng/dL; IPSS < 8	Randomization (1:1:1) to T enanthate 200 mg every two weeks + P orally, or T enanthate 200 mg every two weeks + finasteride 5 mg/day orally or P alone (injection + pill) for 36 months	Mean change in Digit Span Memory Test; Benton Judgment of Line Orientation test and Benton Visual Retention test (working memory); Selective Reminding Test (verbal memory); Beck Depression Inventory (depression); Spielberger State–Trait Anxiety Questionnaire.	No clinically significant effect on tests of cognitive function.
Gray et al. (2005) [70]	Healthy men; 60–75 years (mean 67 years); low-normal gonadal status; AUA symptom score < 7	60 men randomized to 5 groups: 1. T enanthate 25 mg a week (*n* = 13); 2. T enanthate 50 mg a week (*n* = 12); 3. T enanthate 125 mg a week (*n* = 12); 4. T enanthate 300 mg a week (*n* = 13); 5. T enanthate 600 mg a week (*n* = 10) for 20 weeks	Data set analysis: 1. Allocated 13, completed 13, and analyzed 10; 2. Allocated 12, completed 12, and analyzed 11; 3. Allocated 12, completed 11, and analyzed 9; 4. Allocated 13, completed 10, and analyzed 8; 5. Allocated 10, completed 6, and analyzed 6; mean change in sexual outcomes (number of erections, masturbation, sexual activity, etc.); Hamilton’s Depression Rating Scale; Young’s Mania Scale; a computer-based checkerboard visual–spatial test.	No changed in mood were found. A slight but significative improvement in visual–spatial cognition was observed regardless of T posology.
Cherrier et al. (2005) [71]	Healthy men; age 50–90 years (mean 65 years); no cognitive impairment normalized for age (Mattis Dementia Rating Score > 130)	61 men randomized (1:1:1) to: 1. T enanthate 100 mg a week i.m. + anastrozole 1 mg each day; 2. T enanthate 100 mg a week i.m. + P pill each day; 3. P only (injection + pills) for 6 weeks	Mean change at 3, 6, and 12 weeks in route test (spatial memory); story recall (verbal memory); Self-Ordered Pointing Test (working memory); Stroop Color–Word Interference Task (selective attention); verbal fluency.	Spatial memory improved in groups 1 and 2. Verbal memory improved only in group 1. Authors concluded that verbal memory improvement, induced by T administration, depended on aromatization of T to estradiol, whereas improvement in spatial memory occurred in the absence of increases in estradiol.
Haren et al. (2005) [72]	Healthy men; age > 60 years (mean 68.5 years); total T > 230 ng/dL; free testosterone index of 0.3–0.5 (low-to-normal T concentrations)	76 men randomized (1:1) to T undecanoate 80 mg twice a day, orally, vs. P for 12 months	Mean change in Trail-Making Test (Part B), Visuospatial Block Design Test, MMSE, GDS, a 5-Point Likert and a 10-Point Visual Analogue Quality of Life Scale, along with serum hormone measurements. Data were obtained at baseline, 6, and 12 months.	No difference in scores between the two groups.
Bashin et al. (2005) [73]	Low-normal gonadal and healthy men; mean age 66 years	60 older (60–75 years) and 61 youngers (19–35 years); all patients received GnRH agonist to suppress endogenous T production and were finally randomized to: 1. T enanthate 25 mg/weekly (*n* = 13); 2. T enanthate 50 mg/weekly (*n* = 12); 3. T enanthate 125 mg/weekly (*n* = 12); 4. T enanthate 300 mg/weekly (*n* = 14); 5. T enanthate 600 mg/weekly (*n* = 10). Study protocol: 4 weeks control period; 20 weeks of active treatment; 16 weeks recovery	Mean change in fat-free mass, fat mass, muscle strength, sexual function, mood (Hamilton’s Depression Rating Scale; Young’s Mania Scale), visuospatial cognition (a computer-based checkerboard visual–spatial test), hormone concentrations, and safety measures were evaluated before, during, and after treatment of 60 older men who were randomized, 52 completed the study.	Mood and visuospatial cognition did not change significantly in either group.
Kenny et al. (2004) [74]	Age 73–87 years (mean age 80 years); bioavailable T concentrations < 128 ng/dL; mild-to-moderate cognitive impairment (MMSE score 14–28)	11 men randomized (1:1) to T enanthate 200 mg every 3 weeks vs. P for 12 weeks	Mean change in sex hormones (testosterone, bioavailable testosterone, sex hormone binding globulin, estradiol, and estrone), Behave AD Questionnaire (aggressive behavior), Katz Activities of Daily Living (behavior), GDS (mood), Digit Span (attention and recognition), Clock Face Drawing (visuo-construction), Clock Face Perception (visuo-perception), Verbal Fluency, Trail-Making B (executive function), IPSS at baseline, 4, and 10 weeks.	This pilot study did not find any modifications in cognitive function.
Cherrier et al. (2004) [75]	Healthy community dwelling volunteers, age 50–80 years	25 men randomized (1:1) to T enanthate 100 mg weekly vs. P (saline infusion) for 6 weeks	Mean change in spatial memory, spatial reasoning, and verbal fluency. Assessment at baseline, 3, and 6 weeks.	Increased serum T concentrations from treatment were positively associated with improvement in spatial reasoning performance, whereas estradiol was associated with a decline in divided attention performance.
Kenny et al. (2002) [76]	Age 65–87 years (mean 76 years); bioavailable T < 128 ng/dL; no cognitive decline	67 men randomized to transdermal T patch 2–2.5 mg every day vs. P for 12 months	Mean change in sex hormones (testosterone, bioavailable testosterone, sex-hormone-binding globulin (SHBG), estradiol, and estrone); Digit Symbol, Digit Span, Trail-Making A and B, Medical Outcome Survey Short-form 36 (health perception); lower extremity muscle strength and power; calcium intake.	34% of patients did not tolerate the treatment. T supplementation induced mild improvement in physical activity and vitality as well as in cognitive flexibility (Tail-Making B).
Cherrier et al. (2001) [77]	Healthy volunteers; age 50–80 years (mean 65 years); no cognitive impairment at baseline	25 men randomized (1:1) to T enanthate 100 mg every week i.m. vs. P for 6 weeks	Mean change in: recall of a walking route (spatial memory); block construction (spatial ability), recall of a short story (verbal memory). Assessment was performed at baseline, 3, and 6 weeks.	Improvement in spatial memory, spatial ability, and verbal memory.
Ly et al. (2001) [77]	Community-dwelling men healthy men; age > 60 years (mean 68 years); T concentrations <433 ng/dL; no baseline cognitive impairment	37 men randomized (1:1) to DHT transdermal gel 70 mg daily vs. P for 3 months	Mean change in modified MMSE: temporal and spatial orientations; registration of three words; attention and calculation/mental reversal; three-word recall; language in five components (naming, repeating, following a three-stage command, reading and obeying, and writing); visual construction (copying two pentagons). Assessment were performed at baseline, at 1, 2, 3, and 4 months.	DHT seems to slightly impair cognitive function according to the results of the modified MMSE.
Wolf et al. (2000) [78]	Healthy men; T concentrations < 350 ng/dL; mean age 69 years; no relevant co-morbidities	30 patients randomized to T enanthate 250 mg i.m. (*n* = 17) or P (*n* = 13).	Mean change in: verbal fluency (many words as possible pronounced in 1 min); spatial memory (memorize a route marked in a city map within 2 min); verbal memory (six-paired words were read to the patient and an immediate and delayed recall was carried out to evaluate the number of remembered words); Stroop and mental rotation. Tests were performed at baseline and 5 days after the randomization.	An acute administration of T did not provide any relevant improvement or deterioration of the explored cognitive functions.

AAMI: Age-Associated Mental Impairment; T: testosterone; P: placebo; AUA: American Urological Association; GDS: Geriatric Depression Scale; MMSE: Mini-Mental State Examination; WAIS-R: Wechsler Adult Intelligence Scale-Revised; PSA: prostate-specific antigen; IPSS: International Prostate Symptom Score; DHT: dihydrotestosterone; SHBG: sex-hormone-binding globulin.

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
