# Peer review of "Age-Related Male Hypogonadism and Cognitive Impairment in the Elderly: Focus on the Effects of Testosterone Replacement Therapy on Cognition"

_geriatrics, 2020, doi:10.3390/geriatrics5040076_

Round 1

Reviewer 1 Report

The authors have reviewed the literature of randomized controlled trials of testosterone replacement in the elderly with the primary objective of changes or improvement in cognition, and recommend further research.

Their reference 17 from the endocrine society guidelines suggest that there is no clinical benefit of testosterone on cognitive function. The authors have reviewed further articles which confirm this conclusion. Therefore there is nothing unique.  They have presented a listing of references but have not delved into why they feel further research is necessary. They have not  added further information such as  grouping the data according  to age, level of testosterone, duration of study, power of the study to show a benefit  or whether the cognitive  tests performed are appropriate. That is how do these studies compare with other studies which show cognitive improvement with drug treatment?

 They left out a critical and thorough meta-analysis of this topic,  Buskbjerg J Endocrine Society 2019 3(8) 1465. The authors need to compare their results with this analysis.

They should modify their manuscript greatly to present a new  assessment.

Also their manuscript is overly detailed to justify 19 pages, and should concentrate on the issue of cognition.  They can easily eliminate lines 43-71, 84-99,105-113,127-164,272-286.           

They should group their data table according to parameter discussed above.

They should compare their data to Buskjerb.

Author Response

Thank you for your comments and suggestions.

Please consider that the number of pages was particularly elevated as they also included an extensive table (4 pages) and 6 pages of references. However, some part of the text has been removed according to your suggestions and authors focalized concisely on the issue of cognition. Therefore, the current number of pages is 16.

It should be noted that it is a narrative review. Thus, the aim of the manuscript is to overview the main trials that assessed the specific theme of the manuscript, and provide critical comments.

However, according to your observations and suggestions, authors re-wrote the discussion and conclusion in light of the very interesting meta-analysis you suggested (Buskbjerg CR, et al. Endocr Soc. 2019;3(8):1465-1484).

Reviewer 2 Report

This review article summarises the result of the studies of testosterone replacement therapy (TRT) on cognitive function in elderly men.  The author conclude that long-term TRT improves specific cognitive function, such as verbal and spatial memory, cognitive flexibility and physical vitality, however, randomized controlled trials did not provide conclusive results and in most cases TRT resulted neither beneficial nor detrimental on cognitive function in elderly men.  The manuscript is very well written and interesting.  I have no major comments, only minor.  These are presented below:

  1. Line 27 the word “disturbs” is used incorrectly.Maybe “decline” would be better?
  2. Please use the word „concentrations” instead of „levels” when referring to serum testosterone.
  3. Line 61: correct “testicle” to “testicular”
  4. Line 107: correct “at morning” to “in the morning”
  5. Line 193: correct “recently started” to “who where prescribed” or “who were put on”
  6. Line 258: attach “100” to “mg” from the line 259 
  7. Line 272: correct “to maintain” to “in maintaining”
  8. Line 336: correct to “final manuscript” to “final version of the manuscript”

Author Response

Thank you for your comments and suggestions. 

All the modifications you suggested have been carried out. Please note that some modifications have been carried-out in order to improve the readability of the text by removing exceeding parts, and for critically specifying some requested aspects.

Round 2

Reviewer 1 Report

The authors have continued to improve in their focus  on cognitive impairment in the elderly.  However, the narrative is hard to follow since it continues to review many aspects of testosterone replacement not relevant to the issue of cognition.   

The introduction is overly long. The results section list 21 trials. The results section did not group or summarize differences in age, dose, duration or testing other than to say that trials were heterogenous.

Line 43 does not address “inevitable” which would mean that reversal of the factors listed would reverse T decline. This was not shown, and lines 43-48 should be removed.

Lines 53 to 63 are confusing or redundant and should be removed.

The sentence lines 68 to74 needs editorial assistance .

Line 80-87 are not pertinent to this discussion

Lines 226-231 should be removed

Lines 232 to 240 requires editorial assistance.

Line 272 should end at line 274 as the data on half-life and cognitive side effects are not demonstrated for transdermal preparations.

Author Response

Thank you for your comments and suggestions.

Authors listed and explicitly commented the main differences among the studies protocols, also providing a critical analysis of the results in relationship to baseline characteristics of participants (mean age of patients, baseline serum T, baseline cognitive status, T formulation dose and exposure). A point-to-point commenting was done, splitting the trials in different clusters according to the studies designs. This section has been allocated in the discussion, and has been highlighted in yellow.

Other modifications of the text have been carried-out as you suggested. Moreover, text was reexamined in order to correct inaccuracies and for ameliorating its readability. If you confirm that editorial assistance is mandatory, please let us known.